# 3′quant mRNA-Seq of Porcine Liver Reveals Alterations in UPR, Acute Phase Response, and Cholesterol and Bile Acid Metabolism in Response to Different Dietary Fats

**DOI:** 10.3390/genes11091087

**Published:** 2020-09-18

**Authors:** Maria Oczkowicz, Tomasz Szmatoła, Małgorzata Świątkiewicz, Anna Koseniuk, Grzegorz Smołucha, Wojciech Witarski, Alicja Wierzbicka

**Affiliations:** 1Department of Animal Molecular Biology, National Research Institute of Animal Production, ul Krakowska 1, 32-083 Balice, Poland; tomasz.szmatola@izoo.krakow.pl (T.S.); anna.koseniuk@izoo.krakow.pl (A.K.); grzegorz.smolucha@izoo.krakow.pl (G.S.); wojciech.witarski@izoo.krakow.pl (W.W.); alicja.wierzbicka@izoo.krakow.pl (A.W.); 2Centre of Experimental and Innovative Medicine, University of Agriculture in Kraków, Al. Mickiewicza 24/28, 30-059 Kraków, Poland; 3Department of Animal Nutrition and Feed Science, National Research Institute of Animal Production, ul Krakowska 1, 32-083 Balice, Poland; m.swiatkiewicz@izoo.krakow.pl

**Keywords:** pigs, fatty acids, 3′quant mRNA-seq, nutrigenomics

## Abstract

Animal fats are considered to be unhealthy, in contrast to vegetable fats, which are rich in unsaturated fatty acids. However, the use of some fats, such as coconut oil, is still controversial. In our experiment, we divided experimental animals (domestic pigs) into three groups differing only in the type of fat used in the diet: group R: rapeseed oil (*n* = 5); group B: beef tallow (*n* = 5); group C: coconut oil (*n* = 6). After transcriptomic analysis of liver samples, we identified 188, 93, and 53 DEGs (differentially expressed genes) in R vs. B, R vs. C, and B vs. C comparisons, respectively. Next, we performed a functional analysis of identified DEGs with String and IPA software. We observed the enrichment of genes engaged in the unfolded protein response (UPR) and the acute phase response among genes upregulated in B compared to R. In contrast, cholesterol biosynthesis and cholesterol efflux enrichments were observed among genes downregulated in B when compared to R. Moreover, activation of the UPR and inhibition of the sirtuin signaling pathway were noted in C when compared to R. The most striking difference in liver transcriptomic response between C and B was the activation of the acute phase response and inhibition of bile acid synthesis in the latest group. Our results suggest that excessive consumption of animal fats leads to the activation of a cascade of mutually propelling processes harmful to the liver: inflammation, UPR, and imbalances in the biosynthesis of cholesterol and bile acids via altered organelle membrane composition. Nevertheless, these studies should be extended with analysis at the level of proteins and their function.

## 1. Introduction

The study of lipid metabolism is becoming increasingly important in the context of the growing incidence of human metabolic diseases like obesity, NAFLD (Non-Alcoholic Fatty Liver Disease), and type 2 diabetes, as well as neurodegenerative disorders and cancer [1,2]. Dietary fats are one of the most critical determinants of the vulnerability of organisms to the development of diseases. Fatty acids are essential components of cell membranes and function as signaling molecules, regulating enzyme activity and preserving homoeostasis [3].

The recent development of advanced sequencing techniques accelerated research on the interactions between nutrients and diet. The RNA-seq results have given insight into the lipid metabolism of many species, including mice, humans, and pigs. The domestic pig is widely used as a model animal in biomedical research. It seems to be perfect for nutrigenomic research since, as an omnivore, it allows researchers to test various diets. Additionally, genetically, it is much more diverse than laboratory rats or mice, and thus better reflects human biological processes. Thus, this species is commonly used as a model in studies of the molecular background of some human diseases, including cancer and atherogenesis [4,5].

Recently, several studies investigated the effect of omega-3 and omega-6 in the diet of pigs on the transcriptome in different tissues [6,7,8], showing that different fatty acid compositions have a significant effect on the expression of genes engaged in fatty acid synthesis and metabolism. We also performed a preliminary study on the impact of different sources of fat in the diet on the liver transcriptome in pigs [9]; however, due to the limited number of samples and identified differentially expressed genes, we were not able to recognize all aspects of the observed effects. In the present paper, we described the comprehensive functional analysis of Quant-seq 3′mRNA-seq quantitative profiling (Lexogen, Vienna, Austria) of liver samples collected from animals receiving rapeseed oil, beef tallow, or coconut oil in the diet.

Although dietary recommendations point to the harmful effects of excessive consumption of saturated fat, there is still a lot of doubt about the health effects of consuming fats from various sources, e.g., coconut or rapeseed oil. Our investigation aimed to identify the possible molecular mechanism of metabolic changes that occur after receiving different fats in the diet and how they contribute to the pathogenesis of diseases.

## 2. Materials and Methods

### 2.1. Animals and Diets

Animals for the study were kept at the Testing Station of the National Research Institute of Animal Production in Grodziec Śląski. The experiment described in this manuscript was conducted on animals used in previously published studies [10,11,12,13]. All procedures included in this study relating to the use of live animals were in agreement with the local Ethics Committee for Experiments with Animals in Cracow (Resolution No. 912 dated 26 April 2012). In this study, we used 16 samples of liver collected from crossbred fatteners divided into three dietary groups, in which the diets differed among each other in terms of fodder fat: 3% rapeseed oil (group R−*n* = 5, 3 gilts and 2 barrows), 3% beef tallow (group B−*n* = 5, 2 gilts and 3 barrows), and 3% coconut oil (group C−*n* = 6, 3 gilts and 3 barrows). Two samples were excluded from the initial number of 18 samples, due to the low quality of RNA-sequencing. All animals were kept in individual straw-bedded pens in uniform conditions. The animals were healthy, and as equal as possible in regards to body weight. The diets of all of the groups were isonitrogenous and isoenergetic (metabolized energy (ME): *R* = 13.4, *B* = 13.3, *C* = 13.4 MJ/kg feed), and were formulated to cover the nutritional requirements of the pigs. The ingredient composition and nutritive value of the diets are presented elsewhere [11], while the fatty acid compositions of the feed mixtures are shown in Figure 1. Briefly, the group I feed mix contained 80% UFA content (44% MUFA and 36% PUFA), group B contained 67% UFA (32% MUFA and 35% PUFA), and group C contained 45% UFA (16% MUFA and 29% PUFA). The experimental fattening lasted from 60 to 118 kg of live weight of the animals. At the end of the experiment, all the pigs were slaughtered by stunning with high-voltage electric tongs (voltage 240–400 V), and samples of subcutaneous adipose tissue from the area between the last thoracic and the first lumbar vertebrae were collected for transcriptome analysis. All samples were stored in a freezer (−85 °C) until analysis.

### 2.2. RNA Isolation and 3′Quant mRNA Library Construction and Sequencing

RNA isolation was performed using the SPLIT RNA Extraction Kit (Lexogen, Vienna, Austria) according to the manufacturer’s recommendations. Quality of RNA was assessed using a Tapestation 2200 (Agilent, Santa Clara, CA, USA), while quantity was evaluated by nanodrop. A sample of 200 ng of RNA was used for library preparation using the QuantSeq 3′ mRNA-Seq Library Prep Kit FWD for Illumina (Lexogen, Vienna, Austria) according to the manufacturer’s protocol. Assessment of library quantity and quality was performed using a Qubit (Thermofisher Scientific, Foster City, CA, USA) and Tapestation 2200 (Agilent, Santa Clara, CA, USA), respectively. The sequencing of the pooled libraries (50 bp single-read) was performed on an Illumina Hiseq 2500 instrument (Illumina, San Diego, CA, USA), using the High-Output v4—SR 50 Cycle kit (Illumina, San Diego, CA, USA) at the DNA Sequencing Center at Brigham Young University (Provo, UT, USA).

### 2.3. Bioinformatic Analysis

Demultiplexed fastq files were downloaded from the sequencing provider server. Next, the quality check, trimming of reads, and mapping of reads were conducted with FastQC 11.8, FLEXBAR 3.5.0, and TopHat 2.1.1 software, respectively. Samtools 1.9, RSeQC, and HTSeq-count 0.11.1 software, and Gtf-Ensembl annotation 96 were used for evaluation of the mapping statistics and read counts. Differential expression analysis was performed using DEseq 2 software. Genes with *p*-adjusted < 0.05 (FDR-False Discovery Rate) Benjamini–Hochberg (BH) adjustment and no fold-change threshold were regarded as differentially expressed. Functional analysis with STRING software was performed separately for upregulated and downregulated genes using the *Sus scrofa* database. Functional analysis with IPA software was performed using porcine gene names with databases for all available species (human, rat, mouse). Figures were made with the Biorender.com and Paint programs.

### 2.4. Quantitative PCR

RNA was reverse transcribed using a high capacity cDNA archive kit (Thermo Fisher Scientific). Next, qPCR was performed using TaqMan gene expression assays and TaqMan gene expression master mix on a QuantStudio 7-flex instrument. Relative quantity data were analyzed on the Thermo Fisher cloud. Statistical significance was assessed using the Relative Quantification application on Thermo Fisher Connect.

## 3. Results

### 3.1. Fatty Acids Profiles of Diets Used in the Experiment

In our experiment, we compared the expression of genes in the liver of pigs fed with three isoenergetic diets differing only in the type of fat. The detailed information about the diets are presented elsewhere [10]. The proportions of the most important fatty acids in each diet are presented in Figure 1. Group R (rapeseed oil) obtained feed with the highest amounts of unsaturated fatty acids (UFA) and the lowest quantity of saturated fatty acids (SFA). The predominant fatty acids in this group were behenic acid (C22:0), eicosapentaenoic acid (EPA) (C20:5), docosahexaenoic acid (DHA) (C22:6), and α-linolenic acid (C18:3). The second group had an intermediate ratio of SFA/UFA and high content of palmitic acid (C16:0), stearic acid (C18:0), arachidonic acid (C20:4), and palmitoleic acid (C16:1). Group C displayed the highest SFA/UFA ratio and a high content of short/medium-chain saturated fatty acids: myristic acid (C14:0), capric acid (C10:0), and lauric acid (C12:0).

Diets used in the study did not change the phenotype of animals dramatically [10]. We observed no differences in the weight of animals, the weight of the liver and main muscles, or backfat thickness. Different diets did not affect average feed utilization or growth of animals. However, substantial differences were observed in the fatty acid composition of the adipose tissue. We found a high correlation between the dietary fatty acid composition of the diet and adipose tissue fatty acid composition [10,11,12].

### 3.2. 3′Quant mRNA Statistics and DEGs Identified in the Liver after Different Dietary Treatments

After 3′quant mRNA-sequencing, we obtained, on average, 2,776,234 raw reads per sample. More than 79% of them were mapped to *Sus scrofa* 11.1. (Table 1). On average, 11,500 genes with read counts >1 were identified in each sample. All gene expression data were submitted to the GEO NCBI database (accession number: GSE144247). Using DESeq2 software, we performed comparisons of gene expression between all three dietary groups (rapeseed oil vs. beef tallow, rapeseed vs. coconut oil, beef tallow vs. coconut oil, the first group in the comparison is the reference group). MA plots and PCA obtained after DESeq2 analysis are presented in Appendix A. We identified 188 DEGs in the group R vs. group B comparison (77 downregulated in group B and 111 upregulated in group B (adjusted *p*-value < 0.05). In the comparison of group R with group C, 93 DEGs were identified (45 downregulated in group C and 48 upregulated in group C). The lowest number of DEGs (53) was noted in the group B versus group C comparison (32 downregulated in group C and 21 upregulated in group C) (Appendix A. Twenty genes with the highest adjusted *p*-value from each comparison are listed in Table 2.

Among the identified DEGs, there were 30 genes in common between comparison groups R vs. B and groups R vs. group C, 25 common genes between comparison groups R vs. B and groups B vs. C and eight genes in common between the group R vs. C comparison and the B vs. C comparison (Figure 2). Two differentially expressed genes (EGFR, HSPA5) were common for all three comparisons. Interestingly, expression of all common genes was altered in the same direction in both groups: in B and C when compared to R, and in R and C when compared to B. All identified genes in common between specific comparisons are listed in Appendix A.

Since there was a sex imbalance in some groups in our study, we performed a comparison between females and males with DEseq2 software, regardless of dietary group. We observed only 11 differentially expressed genes between female and males, and all of them were located on sex chromosomes (Appendix A). What is more, none of the genes differentially expressed between males and females were present among the genes identified as differentially expressed between dietary groups. We concluded that the effect of sex is negligible in our study.

### 3.3. Functional Analysis of Identified DEGs with STRING

To get insights into the biological processes that are activated by specific dietary fats, we performed functional analysis of identified DEGs with String software using *Sus scrofa* genes as a background. We separately analyzed upregulated and downregulated genes in each comparison. We observed 86, 82, and 18 enrichments in rapeseed oil vs. beef tallow, rapeseed oil vs. coconut oil, and the beef tallow vs. coconut oil comparisons, respectively (Appendix A). In R vs. B and R vs. C comparisons, we observed a higher number of enrichments among downregulated genes (Appendix A). Among genes downregulated by beef tallow in the R vs. B comparison, we observed overrepresentation of genes engaged in oxidoreductase activity (*CAT*, *MSMO*, *GLUD1*), cholesterol biosynthesis (e.g., *HSD3B1*, *HMGCS1*, *MVK*), metabolism of steroids (*FDFT1*, *MVK*, *HMGCS1*), and bile acid secretion (*CYP7A1*, *HMGCR*) (Figure 3, Appendix A). Furthermore, among genes upregulated by beef tallow in this comparison, enrichment of genes associated with the metabolism of RNA (*GNL3*, *NOP58*, *NOPB*), cellular response to stress (*HSPA5, HSPA8*, *BAG3*), the urea cycle (*ENSSSCG00000016159*, *SLC25A15*), and metabolism of polyamines (*AMD1*, *ENSSSCG00000016159*, *SLC25A15*) were observed (Figure 4, Appendix A).

In the second comparison (R vs. C), we observed that genes upregulated by coconut oil are overrepresented in pathways connected to stress response and cellular signaling: the regulation of HSF1-mediated heat shock response (*BAG3*, *ENSSSCG00000015140*, *HSPA9*), and NOTCH3 activation and transmission of signal to the nucleus (*EGFR, PSEN2*). Moreover, downregulated genes were enriched in many metabolism-related pathways and terms (e.g., primary metabolic process, organic substance metabolic process, cellular metabolic process, *ANPEP*, *C1QA*, *ERH*, *FTCD*, *HSD3B1*, *MAN1A1*, *RPS3*, *SDHD*, *TMEM86B*, *UOX*).

In the comparison between beef tallow and coconut oil, we observed downregulation of genes engaged in immunity (defense response (*C1QA, HP, IL4R, ITIH4*), and genes engaged in protein processing in the endoplasmic reticulum (*HSPA5, HYOU1, PDIA3*). Genes upregulated by coconut oil were associated with PPAR signaling pathways (*CYP7A1, FABP1)* and bile secretion and cholesterol metabolism (*ABCB11, CYP7A1)*. All enrichments with adequate FDR values and matching proteins are presented in Appendix A.

### 3.4. Functional Analysis of Identified DEGs with IPA

To further analyze differentially expressed gene sets for their association with human diseases and to assess to what extent the relationships observed in pigs can be translated into humans, we performed functional analysis of identified DEGs from all comparisons with ingenuity pathway analysis (IPA) (Qiagen). Sixty-six significant canonical pathways were observed in the comparison between group R vs. B (log2 *p*-value > 1.5) (Appendix A). Among them, the superpathway of cholesterol biosynthesis (Z-score = −2.646, *p*-value < 2.34 × 10^9^, mevalonate pathway I (Z-score = −2, *p*-value < 3.57 × 10^6^), the superpathway of geranylgeranyldiphosphate biosynthesis I (Z-score = −2, *p*-value < 1.07 × 10^5^), and LXR/RXR activation (Z-score = −0.447, *p*-value < 5.12 × 10^5^) were inhibited, while LPS/IL-1-mediated inhibition of RXR function (Z-score = 1.342, *p*-value < 1.25 × 10^5^), acute phase response signaling (Z-score = 2.236, *p*-value < 5.74 × 10^4^), Huntington`s disease signaling (Z-score = 1, *p*-value < 2.88 × 10^3^), and several pathways of inositol metabolism were activated by the beef tallow diet when compared to the rapeseed oil diet (Figure 5) (Appendix A). In the second comparison (rapeseed oil vs. coconut oil), 30 canonical pathways were noted, with the sirtulin signaling pathway (Z-score = −2, *p*-value < 1.1 × 10^4^) being inhibited and the BAG2 signaling pathway (Z-score = 1, *p*-value < 2.01 × 10^5^) and unfolded protein response (Z-score = 2, *p*-value < 5.76 × 10^5^) being activated by coconut oil (Figure 5). The beef tallow vs. coconut oil comparison revealed 43 enriched canonical pathways. Beef tallow activated acute phase response signaling (Z-score = −2.236, *p*-value < 4.32 × 10^9^), while coconut oil stimulated RXR/LXR activation (Z-score = 0.816, *p*-value < 7.24 × 10^9^) in this comparison (Figure 5). All identified canonical pathways and connections between them are presented as networks in Appendix A.

### 3.5. Identification of Hub Genes with Cytohubba

Our next step was to identify critical genes responsible for changes observed in the transcriptome under the influence of different types of fat. For this purpose, we used the cytoHubba–Cytoscape plugin for ranking nodes in a network by their network features. Among eleven available methods, we chose MCC, which is the most precise in predicting essential proteins from the yeast PPI network [13]. In the first comparison, R vs. B, *GNL3* was ranked number one, followed by *RSL1D1, UTP3*, and *DDX24*. The essential genes in the B vs. C comparison were *ORM1* and *TTR*, while in the rapeseed oil vs. coconut oil they were *RPS3* and *HSPA9* (Figure 6).

### 3.6. Validation of Quant 3′mRNA Profiling by qPCR

To validate the 3′quant mRNA-seq results, we performed qPCR analysis of several DEGs identified in the study. We observed high concordance between RNA-seq and qPCR results. As expected, we found strong overexpression of genes engaged in the unfolded protein response (*BAG3*, *HSPA5*) and inflammation (*PID1*, *IHIT4*, *ALPL*, *LITAF*) in the beef tallow group compared to the rapeseed oil and coconut oil groups. In contrast, genes involved in bile acid secretion (*CYP7A1*) and cholesterol biosynthesis *(HSD3B1)* were downregulated in the beef tallow group when compared to both groups or the rapeseed oil group only (Figure 7). Furthermore, none of the genes analyzed by qPCR showed significantly different expression between gilts and barrows.

## 4. Discussion

The progress of civilization has initiated many changes in the human environment. This applies not only to nature that surrounds us but also to our way of life, especially our nutrition. First of all, fat consumption and the amount of saturated fatty acids consumed increased significantly during the last century. Moreover, the ratio of omega-6 to omega-3 in food decreased, resulting in worsening of the dietary fatty acid profile [14]. At the same time, there was a sharp increase in the incidence of diseases related to metabolism, cardiovascular diseases, neurodegenerative diseases, and cancer. These diseases are also called civilization diseases because their occurrence is closely related to the modern lifestyle.

In our experiment, we used a domestic pig (*Sus scrofa*) to illustrate the changes that occur in the liver transcriptome as a result of using various fats in the diet. Using isoenergetic and isonitrogenous diets made it possible to observe transcriptome changes at a specific stage, before drastic changes at the phenotype level occurred. For quantitative analysis of the transcriptome, we chose the 3′quant mRNA method, which allows transcriptome profiling based on the 3′UTR ends of the gene and is a cost-efficient alternative to whole transcriptome RNA-seq. By this method, we identified 11,500 genes with >1 read number, of which 308 (2.67%) were differentially expressed between samples of animals fed different diets, despite a very low sequencing coverage—approximately two million reads per sample. We observed that many of these genes are engaged in the pathogenesis of human civilization diseases and that depending on the source of dietary fat, different pathways are activated or inhibited in liver at the gene expression level.

### 4.1. Biosynthesis and Catabolism of Cholesterol are Inhibited in Animals Obtaining Beef Tallow in the Diet

The opinion that the consumption of saturated fats increases blood cholesterol levels is generally approved and well documented [15]. Maintaining adequate blood cholesterol levels is crucial for the body since its excess results in the development of atherosclerosis, heart disease, and neurodegenerative diseases. On the other hand, cholesterol is an essential component of cell membranes and has many important functions in the organism. Cholesterol homeostasis in the body is directed by the interaction between absorption, synthesis, and excretion or conversion of cholesterol into bile acids. A reciprocal relationship between these processes is known to regulate circulating cholesterol levels in response to dietary or therapeutic interventions. Cholesterol biosynthesis is self-regulating; a high cholesterol level in the blood forces the inhibition of its synthesis in the liver. One of the mechanisms involved in this process is inhibition of the expression of the gene encoding HMGCR, a rate-limiting enzyme in cholesterol biosynthesis. This is exactly what we observed in animals fed beef tallow—the expression of *HMGCR* was ~3 fold lower than in animals fed rapeseed oil. We observed inhibition of several other genes engaged in cholesterol biosynthesis (*FDFT1*, *MVD*, *EBP*, *HSDB31*) in the group receiving beef tallow in the diet when compared to the group receiving rapeseed oil. Two of these genes (*FDFT1*, *HSD3B1*) were also downregulated in the coconut oil group compared to the rapeseed oil group but to a lesser extent, suggesting a different response of the mechanisms responsible for cholesterol homeostasis to medium-chain saturated fatty acids. The results of the functional analysis using the STRING and IPA programs indicate the existence of an additional mechanism affecting the level of cholesterol synthesis. According to the IPA results (Figure 5), RXR function was inhibited through LPS/IL-1 mediation, and simultaneously the super pathway of cholesterol biosynthesis was repressed in the beef tallow group. We suppose that it may have occurred as a result of a series of changes triggered by gut microbiome misbalance under the influence of dietary beef tallow. A diet rich in saturated fatty acid affects gut microbiota composition by enhancing overflow of dietary fat to the distal intestine in mice [15]. In the pigs fed beef tallow, an increase in pathogenic LPS-secreting bacteria appeared, which resulted in an increase in expression of *LBP* and other acute-phase response genes in the liver. Moreover, beef tallow contains substantial amounts of arachidonic acid, which is known for its pro-inflammatory properties. As a consequence of inflammation, the RXR function was inhibited, which affected the level of expression of genes engaged in cholesterol metabolism (Figure 8).

The use of coconut oil in the diet still causes a lot of controversies. As a fat containing many saturated fatty acids, and thus causing an increase in blood cholesterol levels, it is not recommended for people at risk of cardiovascular disease. On the other hand, it was shown recently that it may improve intestinal microbiota, antioxidant status, and immunity of growing rabbits [16]. Moreover, the latest meta-analysis showed that MCFA (medium-chain fatty acids), which predominate in coconut oil, increase HDLcholesterol—responsible for cholesterol efflux—content in comparison with long-chain fatty acids. *CYP7A1* is a gene coding for a rate limiting enzyme in the cholesterol catabolic pathway in the liver, which converts cholesterol to bile acids. This reaction is the major site of regulation of bile acid synthesis, which is the primary mechanism for the removal of cholesterol from the body. We observed a 6-fold decrease in *CYP7A1* gene expression in the beef tallow group compared to coconut oil, emphasizing the advantage of coconut oil over beef tallow in cholesterol efflux. It was shown in vitro that arachidonic acid is a potent inhibitor of *CYP7A1* expression [17], which is in accordance with our results.

### 4.2. Acute-Phase Response Signaling is Activated in the Beef Tallow Group when Compared to both Rapeseed Oil and Coconut Oil

Low-grade inflammation is one of the leading causes of NAFLD, cardiovascular diseases, diabetes, and neurodegenerative diseases. We observed a significant increase in the expression level of many genes connected to the immune system response and inflammation markers in liver tissue collected from animals fed with beef tallow (*LITAF*, *ALPL*, *PID1*, *IHIT4*, *LBP*, *HP*) (Figure 7, Appendix A). Strikingly, the *LBP* gene, which codes for lipopolysaccharide binding protein, and *CD14* were both upregulated in the beef tallow group. LBP and CD14 drive ternary complex formation and TLR activation and as a consequence trigger the whole cascade of immune response stimulated by *NFKBiB* [18]. Interestingly, the expression of some genes (*LBP*, *IHIT4*) was also upregulated in rapeseed oil when compared to coconut oil, supporting information about the antibacterial properties of coconut oil [16]. Ingenuity pathway analysis revealed that the acute phase response canonical pathway is highly significantly activated in animals obtaining beef tallow when compared to both rapeseed oil and coconut oil. The pro-inflammatory effects of long-chain saturated fatty acids have been known for some time [19]. It has been observed that long-chain saturated fatty acids increase haptoglobin gene expression (inflammation marker) in mice adipose tissue [19]. Another study showed that the composition and metabolic activity of the gut microbiota change as a result of a steatohepatitis-inducing high-fat diet in mice [20]. Moreover, in these animals, the level of saturated fatty acids (palmitic acid) in the gut increased significantly, activated macrophages in the liver, and promoted TNF-α expression. The consequence of these changes was the development of NASH, which was reversible under the influence of antibiotics [20]. On the other hand, lauric acid—the main component of coconut oil—was shown previously to alleviate neuroinflammatory responses by LPS-activated microglia, supporting its beneficial effect on neurodegenerative diseases [21]. Our results, in agreement with previous studies, underline the difference in response of the immune system to dietary long-chain SFA (beef tallow) and medium-chain SFA (coconut oil), (Figure 9).

### 4.3. Unfolded Protein Response is Activated in both the Beef Tallow and Coconut Oil Groups when Compared to the Rapeseed Oil Group

The unfolded protein response (UPR) is a highly conserved pathway that allows the cell to manage endoplasmic reticulum (ER) stress that is imposed by the secretory demands associated with environmental forces [22]. Our results show that upon dietary beef tallow or coconut oil uptake, expression of genes engaged in UPR—*BAG3*, *HSP5*, *HSP7*—increases when compared to rapeseed oil. Consequently, functional analysis with IPA and STRING revealed that UPR is strongly activated in the coconut oil and beef tallow groups when compared to rapeseed oil. qPCR analysis confirmed overexpression of *HSPA5* and *BAG3* in beef tallow and both saturated fatty acids rich groups, respectively, when compared to rapeseed oil. All these results support the opinion that UPR is regulated by lipid-dependent mechanisms [23,24]. It is assumed that in an environment rich in saturated fatty acids, the composition of membranes in the endoplasmic reticulum changes, leads to disturbances in protein folding and results in activation of the unfolded protein response pathway [24]. Recent in vitro studies [25] indicate that palmitic acid induces ER stress and simultaneously increases inflammatory indices, while oleic acid ameliorates this action in exocrine pancreas cells. In our experiment, the ratio of palmitic to oleic acid was about twice as high in coconut oil and beef tallow as compared to rapeseed oil (Figure 1), supporting in vivo associations previously observed in vitro in pancreas cells [25].

When the proteins present in mitochondria are damaged, and their accumulation threatens the maintenance of homeostasis, mitochondrial specific UPR (UPR^mt^) is activated. The central UPR^mt^ coordinator is *SIRT3*, which encodes a member of the sirtuin family of class III histone deacetylases. In our study, we observed lower expression of *SIRT3* in animals fed coconut oil in the diet when compared to the rapeseed oil group. Moreover, according to IPA analysis, the sirtuin signaling pathway (with *SIRT3*, *SDHD*, *UOX* genes) was downregulated in the coconut oil group. This result may have an important clinical significance since the loss of *SIRT3* leads to deregulation of several mitochondrial pathways, which contributes to the accelerated development of the disease of ageing [26]. In general, lowering the *SIRT3* level is associated with adverse health effects. It is considered a mitochondria-localized tumor suppressor, which opposes reprogramming of cancer cell metabolism through HIF1α destabilization [27]. Moreover, it was shown that *SIRT3* deficiency accelerates the development of metabolic syndrome. On the other hand, in vivo experiments show that a chronic high-fat diet decreases expression of *SIRT3* in liver tissue [28]. In contrast, in vitro experiments suggest that palmitic acids increase *SIRT3* expression, contrary to oleic acids [29]. Thus, the effect of coconut oil consumption on the sirtuin signaling pathway should be further analyzed on the protein and protein function levels, especially in the context of using it to prevent neurodegenerative diseases.

During UPR activation, a decrease in RNA synthesis is often observed to protect cells against excessive accumulation of misfolded proteins [30]. According to our results, dietary beef tallow-activated genes (*DDX21*, *NOLC1*) suppress pre-rRNA transcription by forming ring-shaped structures surrounding multiple Pol I complexes [31,32]. Additionally, we observed upregulation of *NOB1*, which blocks the recruitment of mRNAs to the nascent ribosome [33]. Thus, it suggests the launching of repair mechanisms to stop the excessive production of misfolded proteins in the beef tallow group. On the other hand, several genes (*EIF3A*, *EIF5B*) engaged in translation initiation were upregulated in the beef tallow and coconut groups. It was recently found that eIF3a regulates HIF1α protein synthesis through the internal ribosomal entry site (IRES)-dependent translation. Therefore, it was concluded that eIF3a might be a potential therapeutic target for hepatic carcinoma (HCC) since it acts as a regulator for glycolysis—a process that is central to cancerous reprogramming of metabolism [34].

### 4.4. UPR, Inflammation, and Cholesterol and Bile Acid Metabolism are the Main Processes Affected by Dietary Fatty Acids

In our experiment, we observed that UPR, inflammation, and cholesterol and bile acid metabolism are the most altered processes under different dietary fat treatments. Interaction between endoplasmic reticulum stress—a trigger for UPR and inflammation—is involved in a variety of human pathologies [22,35]. Our results show that cholesterol and bile acid metabolism are additional components of this complex. They also support the hypothesis that a key point in these interactions is the composition of lipid membranes, since “organelle membrane” is the common cellular component enriched in all comparisons of our study (Appendix A). Considering the results of functional analysis of the identified DEGs as well as the qPCR results, it can be stated that the UPR level is low in the rapeseed oil group, while it is high in the beef tallow and coconut oil groups. The level of inflammation and cholesterol efflux is high only in the group receiving beef tallow. In contrast, cholesterol biosynthesis is low in the group receiving beef tallow and to a lesser extent in the coconut oil group. Our next step was the analysis of the causal relations between these processes and the identification of key genes regulating these interactions. The effect of bile acid and cholesterol metabolism on inflammatory processes has been well known for a long time. Numerous studies indicate that reduced flow of bile acids leads to their accumulation in the liver, which in turn causes inflammation. It was shown that bile acids act as an inflammagen, and directly activate signaling pathways in hepatocytes that stimulate the production of the proinflammatory mediators. However, other mechanisms were considered; the inflammatory response is triggered by activation of Toll-like receptor 4 (TLR4), either by bacterial lipopolysaccharide or by damage-associated molecular pattern molecules released from dead hepatocytes [36]. When we compared the effect of beef tallow and coconut oil in relation to rapeseed oil, we observed the downregulation of genes responsible for the synthesis and transport of bile acids (*CYP7A1*, *ABCB11*) only in the group receiving beef tallow. Similarly, only in this group, we observed an increase in expression of genes coding for acute-phase proteins. The exception here was the *ORM1* gene (identified as a hub gene by cytohubba software), which was upregulated in the coconut oil group. The product of this gene is classified as an acute phase protein, but it also has immunosuppressive activity. More than 25 years ago, it was found that this protein protects mice against lethality shock induced by tumor necrosis factor (TNF) or endotoxins [37]. More recently, a decrease of ORMDL protein without decreases in ORMDL mRNA levels was observed in HepG2 liver cells treated with the pro-inflammatory stimulus, and this observation was extended to in vivo models of inflammation [38]. In contrast, we observed a decrease in ORM expression accompanied by an increase in acute phase response in the beef tallow group at the mRNA level. The situation is further complicated by the fact that the *ORM1* gene is activated by bile acid through FXR—the nuclear bile acid receptor in mice [39]. This may indicate that the main difference in the action of coconut oil compared to beef tallow is that coconut oil does not reduce cholesterol catabolism and its disposal with bile acids. Thanks to this, FXR is activated, and thus ORM1′s protective effect is maintained.

The interaction between inflammation and cholesterol and bile acid metabolism also acts in the opposite direction; the activation of immune response proteins affects the level of gene expression associated with cholesterol and bile acid metabolism [40]. In the liver, LPS markedly decreases the mRNA expression and activity of CYP7A1 and cholesterol transporters ABCG5 and ABCG8, which mediate cholesterol excretion into the bile in Syrian hamsters [41]. The relationship discussed here has a significant clinical implication. Until recently, hypercholesterolemia was considered the main cause of heart disease, but some scientists indicate that inflammation may play a more important role in the pathogenesis of CVD. Our results underline the significance of inflammation in this context. In essence, findings from epidemiological studies report low rates of cardiovascular disease among populations who consume coconut oil as part of their traditional diets (in India, the Philippines, and Polynesia) even though this fat is cholesterol-raising. Considering that in our study the animals receiving coconut oil did not show an increase in activation of acute-phase response, the limited influence of coconut oil consumption for the occurrence of heart disease may be due to the anti-inflammatory properties of this oil (Figure 9).

Inflammatory processes are strictly connected to endoplasmic reticulum stress, and UPR and the *ORM* genes link these two processes. Yeast cells lacking the *ORM1* and *ORM2* genes show a constitutive unfolded protein response, sensitivity to stress, and slow ER-to-Golgi transport [42]. ORM proteins function as the main regulators of the enzyme serine palmitoyltransferase—necessary for the process of sphingolipid biosynthesis, which is one of the main components of the cytoplasmic membranes of the endoplasmic reticulum, and disturbances in its homeostasis lead to ER stress [43]. The unfolded protein response can also alter sphingolipid metabolism. Bi-directional interactions between sphingolipids and the UPR have now been observed in a range of diseases, including cancer, diabetes, and liver disease [43]. Other studies have shown that ER stress can directly initiate the proinflammatory pathways, while proinflammatory agents such as ROS, TLR ligands, and cytokines can induce ER stress. As a result, activated UPR may further enhance the pro-inflammatory response [44].

As in the case of relationships discussed earlier, the connection between UPR and biosynthesis of cholesterol and bile acids is well documented and mutual. It was shown that within the ER, there are numerous membrane receptors detecting changes in cholesterol levels and cholesterol overload causes severe dysregulation of the ER [45]. To counteract these abnormalities, transcription factors (e.g., *SREBF2*, *LXR*) regulating cholesterol biosynthesis and immunological responses are triggered by the NRF2 protein [46]. In our experiment, we did not observe changes in the level of expression of genes coding for these molecules; however, they are largely regulated at the level of translation or post-translational modifications [47]. Among the DEGs identified in our report, the two genes *S1PR1* (upregulated in the beef tallow and coconut oil groups when compared to the rapeseed oil group) and *ORAI3* (downregulated only in the beef tallow group when compared to the rapeseed oil group) are potential links between cholesterol metabolism and ER stress and UPR. The protein encoded by *S1PR1* is structurally similar to G protein-coupled receptors and binds the ligand sphingosine-1-phosphate (S1P) with high affinity and high specificity. S1P, in turn, is a signaling sphingolipid acting as a bioactive lipid mediator. It is transported mainly by HDL and activates one of the S1PR1-mediated biological functions: calcium flux [48]. It was recently demonstrated that S1PR1-mediated calcium efflux is achieved through ORAI-membrane calcium channels. Thus, our data suggest that both the dietary fats beef tallow and coconut oil activate *S1PR1*, but only beef tallow downregulates *ORAI3* transcript expression. Interestingly, store-operated calcium (Ca^2+^) entry (SOCE) is mediated by *Orai3* only in breast cancer cells that express the estrogen receptor, contrary to estrogen receptor-negative cancer cells, which suggests a relationship between estrogen concentration and *ORAI3* expression [49]. In our experiment, probably as a consequence of reduced cholesterol catabolism in the beef tallow group, we observed a decrease in steroid hormone biosynthesis, which theoretically could lead to a decrease in *ORAI3* expression and dysfunctional calcium channels.

Due to the limited amount of space, we are not able to discuss all the interesting relationships observed in our experiment. To mention only a few: the “inositol metabolism”, “urea cycle”, and “glutathione metabolism” pathways deserve additional detailed analysis in terms of the effect of dietary fats on cancer and the development of metabolic, neurodegenerative, and cardiovascular diseases. Moreover, expression of a number genes—potential therapeutic targets in liver diseases—including *EGFR*, *GLUD1*, *GNL3*, and *RGS5*, has been shown to be modified by dietary fats and should be further investigated in this regard. Although our work provides a large amount of new information on the impact of consuming different sources of fat on gene expression in the liver, we are aware that these studies should be extended with analysis at the level of proteins and their function. Furthermore, analysis of bile acid species content in the intestine and liver and histological examination of these organs would give a full view of changes introduced by consumption of different sources of fat. Even though the material was collected from all animals from the same part of the liver, our samples most likely contained a mixture of different cells (hepatocytes, parenchymal cells, immune cells). Therefore, differences in expression observed between the groups of animals tested may partly result from the proportion of the content of these cells in the samples. To accurately assess what processes occur in the specific cells of the liver under the influence of dietary fat, experiments using single-cell methods are necessary.

## 5. Conclusions

To sum up, our experiment showed that the type of consumed fat has a very significant impact on changes in gene expression in the liver of pigs. We observed that these changes are most intensively visible in three related processes: cholesterol and bile acid biosynthesis, UPR, and inflammation, playing a key role in the pathogenesis of civilization diseases. If one of these processes is dysregulated, repair mechanisms are triggered by activation of connected pathways. Therefore, when the body is excessively stimulated by improper nutrition, a vicious circle starts, in which the dysregulation of one process results in the dysregulation of the next. In our experiment, this situation most likely arose as a result of a diet containing beef tallow. In this group of animals, we observed deregulation of cholesterol and bile acid metabolism, activation of genes coding for acute phase proteins, and activation of the UPR when compared to animals fed with rapeseed oil. In contrast, in the coconut oil group, no activation of inflammation genes was observed, suggesting that some ingredients of coconut oil (lauric acid, polyphenols) can stop this vicious cycle and prevent the development of civilization diseases.

## Figures and Tables

**Figure 1 genes-11-01087-f001:**
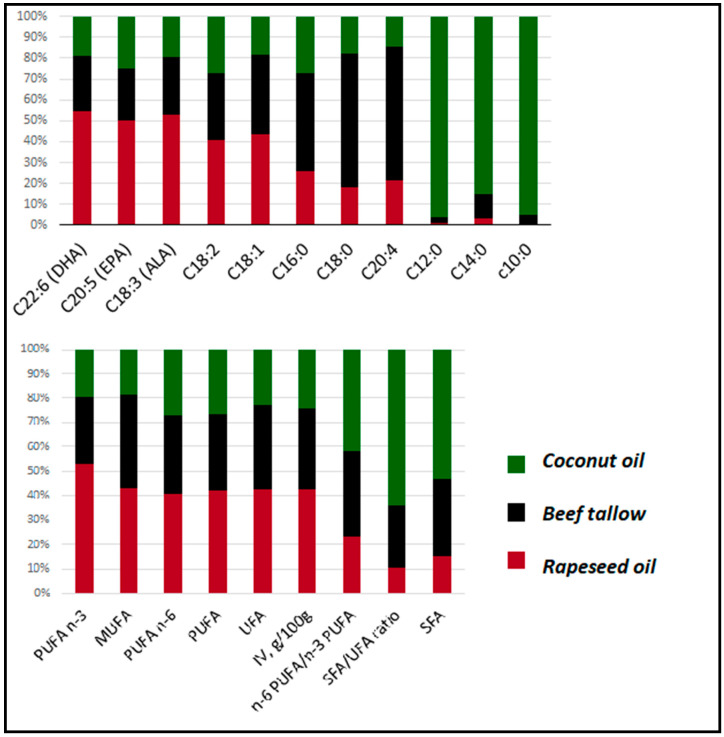
Percentage of the most abundant fatty acids in the diets.

**Figure 2 genes-11-01087-f002:**
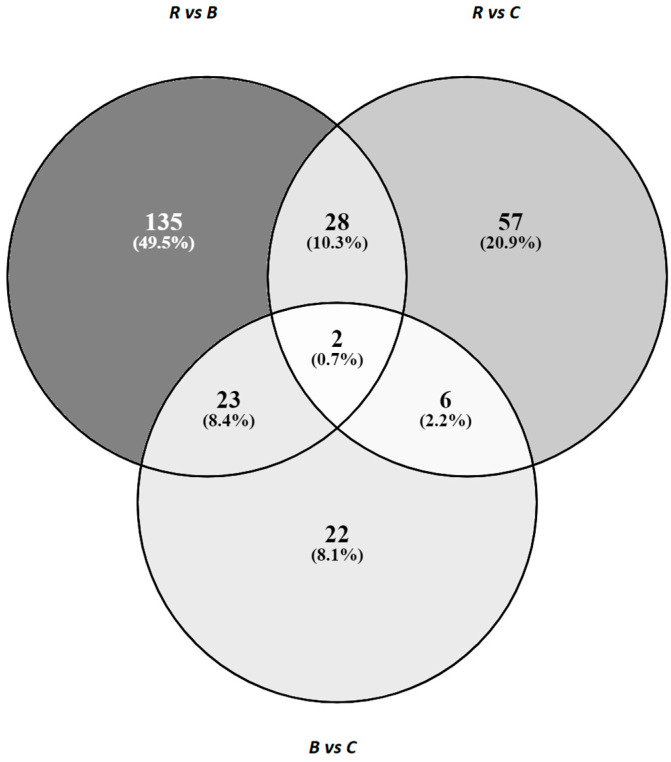
Venn diagram illustrating differentially expressed genes that are common between comparisons.

**Figure 3 genes-11-01087-f003:**
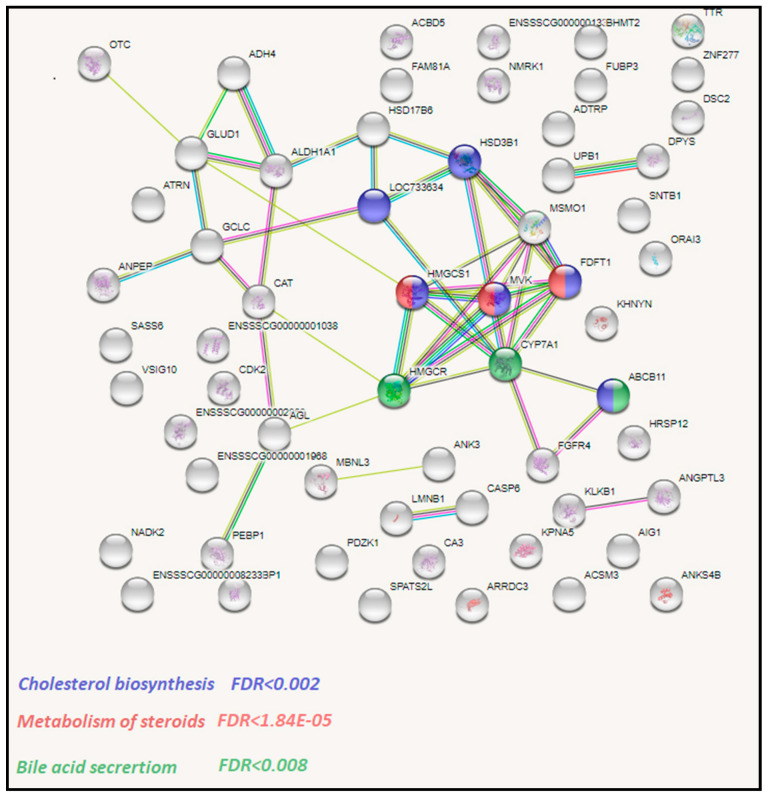
The network of DEGs downregulated in the beef tallow group when compared to the rapeseed oil group according to STRING software analysis.

**Figure 4 genes-11-01087-f004:**
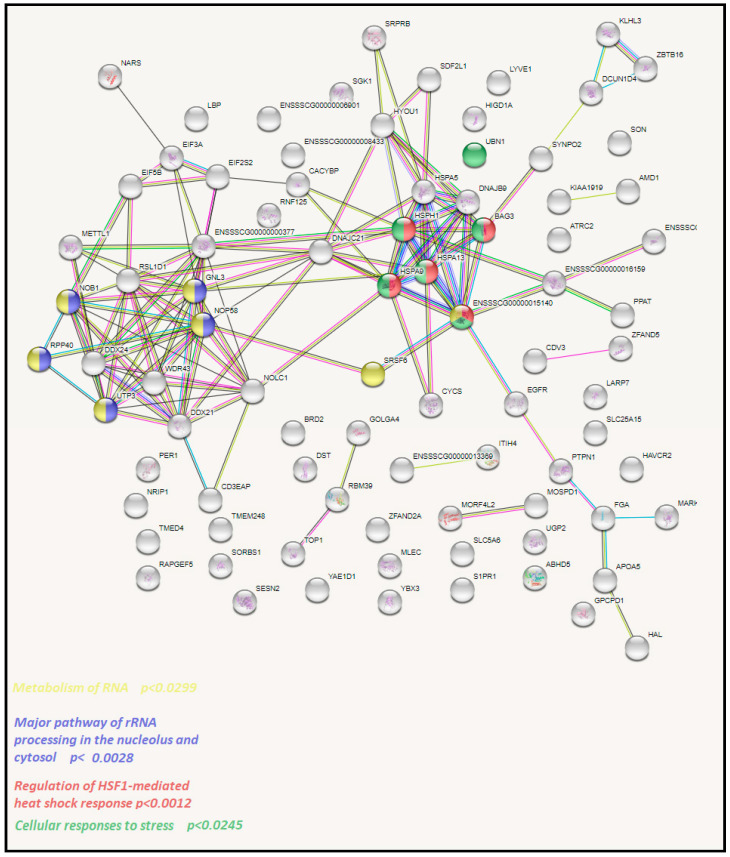
The network of DEGs upregulated in the beef tallow group when compared to the rapeseed oil group according to STRING software analysis.

**Figure 5 genes-11-01087-f005:**
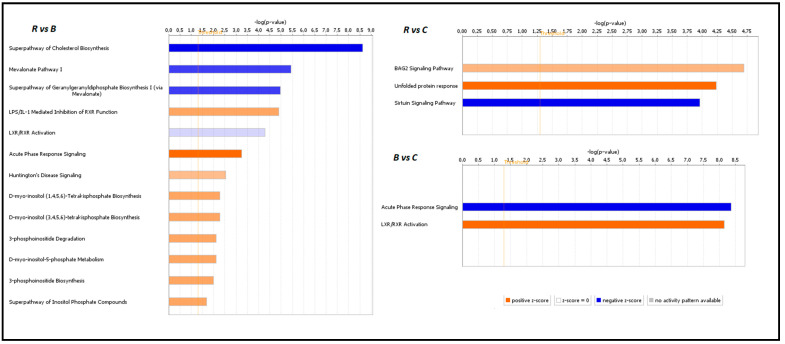
Canonical pathways with Z score ≠ 0 and log (*p*-value) > 1.3, identified in each comparison using IPA software.

**Figure 6 genes-11-01087-f006:**
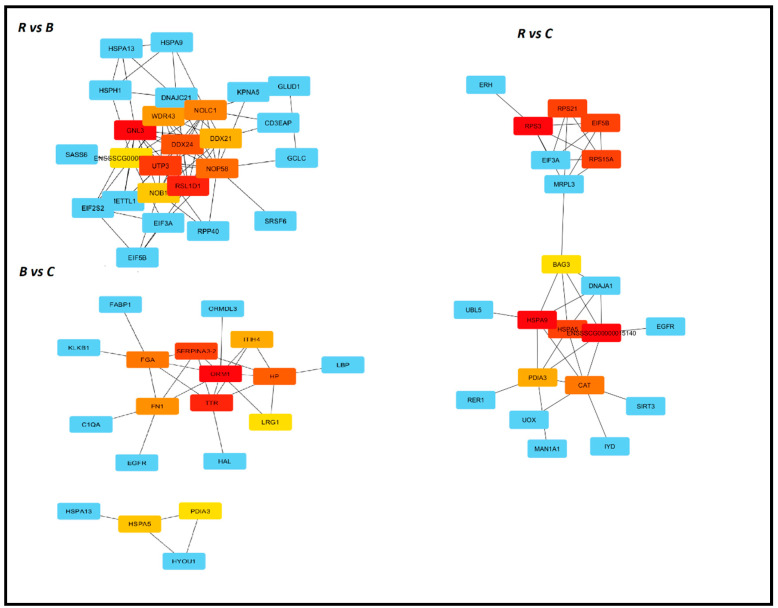
Top 10 Hub genes identified in the R vs. B comparison, R vs. C comparison, and B vs. C comparison using Cytohubba, ranked by MCC (Maximal Clique Centrality). The more intense the red color, the higher the position in the rank.

**Figure 7 genes-11-01087-f007:**
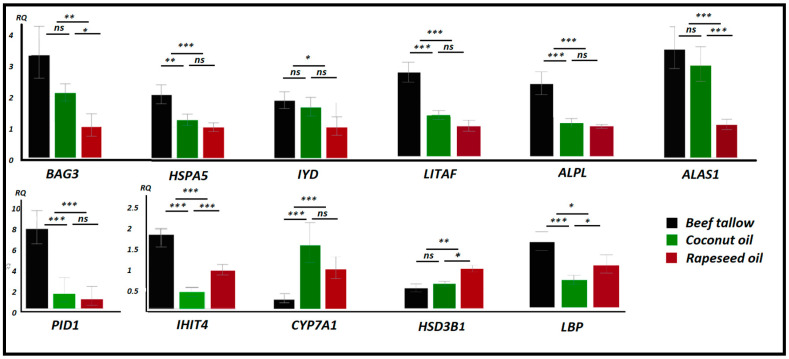
Results of the qPCR analysis of selected DEGs. RQ—Relative Quantity of mRNA, *** *p* < 0.01, ** *p* < 0.05, * *p* < 0.1, ns—not significant.

**Figure 8 genes-11-01087-f008:**
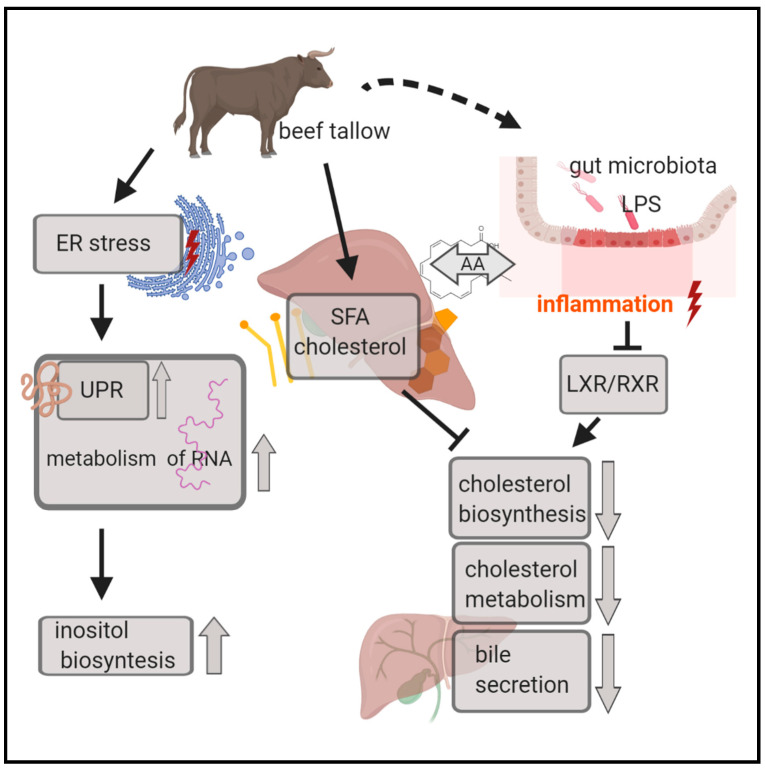
Graphical illustration of selected biological processes and canonical pathways altered in the beef tallow group when compared to rapeseed oil. Beef tallow contains pro-inflammatory ingredients (SFA, AA—arachidonic acid) that could change gut microbiota and promote inflammation. SFA and cholesterol from beef tallow decrease cholesterol biosynthesis directly in the liver and indirectly through inhibition of LXR/RXR by LPS/IL-1.

**Figure 9 genes-11-01087-f009:**
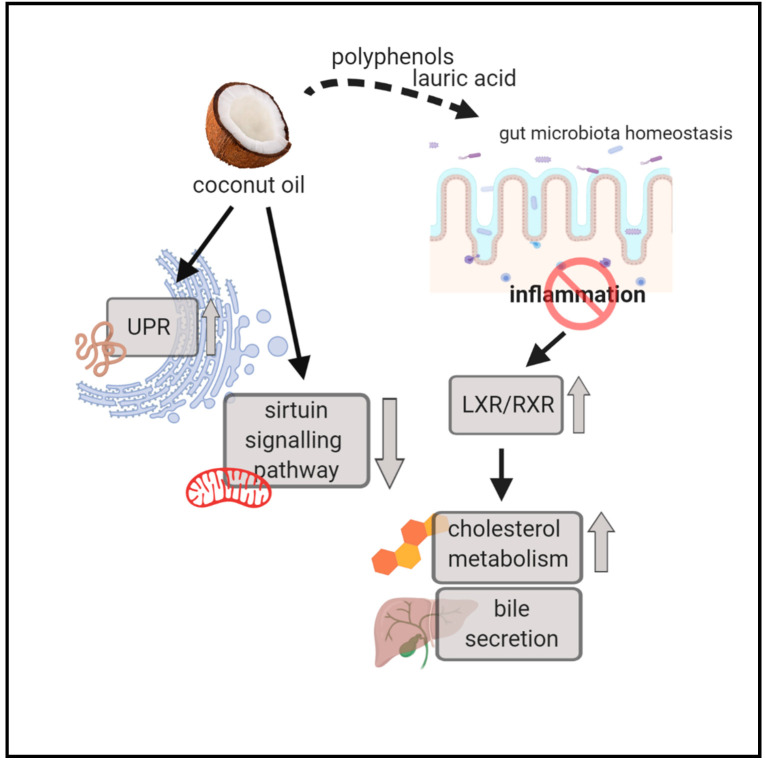
Graphical illustration of selected biological processes and canonical pathways altered in the coconut oil group when compared to the rapeseed oil (UPR, sirtuin signaling pathway) and beef tallow groups (LXR/RXR activation, cholesterol metabolism, bile secretion). Coconut oil contains antibacterial and anti-inflammatory ingredients (lauric acids, polyphenols), and cholesterol metabolism and bile acid secretion are not reduced as a result of increased inflammation, which is observed in the beef tallow group.

**Table 1 genes-11-01087-t001:** RNA-seq statistics of Quant-seq 3′mRNA-sequencing.

Sample Name	No of Raw Reads	No of Mapped Reads	Percentage of Mapped Reads (%)
17w	3,355,869	2,635,495	78.5
19w	2,460,130	1,927,026	78.3
20w	2,029,705	1,615,350	79.6
21w	2,819,316	2,187,454	77.6
22w	2,657,611	2,111,384	79.4
24w	2,298,911	1,803,582	78.5
41w	3,469,934	2,760,471	79.6
42w	2,223,698	1,817,882	81.8
43w	4,124,251	3,197,675	77.5
45w	2,157,086	1,710,536	79.3
46w	2,415,641	1,932,621	80.0
48w	2,470,229	1,981,470	80.2
49w	2,715,605	2,203,190	81.1
52w	2,631,687	2,119,487	80.5
53w	2,110,920	1,648,309	78.1
54w	2,031,614	1,625,581	80.0
55w	4,000,000	3,266,253	81.7
56w	4,000,000	3,262,080	81.6

**Table 2 genes-11-01087-t002:** Top 20 DEGs with the highest adjusted *p*-value for each comparison.

Ensembl ID	Base Mean	log2 Fold Change	*p*-Adjusted	Gene Description	Gene Symbol
**Beef Tallow vs. Coconut Oil**					
ENSSSCG00000021443	109	−1.46	0.0005	serum/glucocorticoid regulated kinase 1	*SGK1*
ENSSSCG00000032381	221	−1.42	0.0000	lipopolysaccharide induced TNF factor	*LITAF*
ENSSSCG00000004700	288	−1.34	0.0007	protein disulfide isomerase family A member 3	*PDIA3*
ENSSSCG00000028758	177	−1.28	0.0000	lipopolysaccharide binding protein	*LBP*
ENSSSCG00000000892	127	−1.23	0.0008	histidine ammonia-lyase	*HAL*
ENSSSCG00000033214	460	−1.16	0.0098	glycine N-methyltransferase	*GNMT*
ENSSSCG00000016174	361	−1.16	0.0185	fibronectin 1	*FN1*
ENSSSCG00000008998	23136	−1.01	0.0001	fibrinogen α chain	*FGA*
ENSSSCG00000013514	256	−0.94	0.0001	LRRCT domain-containing protein	***PLIN5***
ENSSSCG00000008550	317	−0.86	0.0124	solute carrier family 5 member 6	*SLC5A6*
ENSSSCG00000022126	122	−0.79	0.0025	epidermal growth factor receptor	*EGFR*
ENSSSCG00000002487	1830	−0.78	0.0027	α-1-antichymotrypsin 2	***SERPINA3-2***
ENSSSCG00000002749	37957	−0.59	0.0162	haptoglobin	*HP*
ENSSSCG00000011453	1755	−1.43	0.0000	inter-α-trypsin inhibitor heavy chain 4	*ITIH4*
ENSSSCG00000011741	340	0.45	0.0185	golgi integral membrane protein 4	*GOLIM4*
ENSSSCG00000027072	172	0.56	0.0112	ATP synthase inhibitory factor subunit 1	*ATP5IF1*
ENSSSCG00000002529	355	0.65	0.0036	40S ribosomal protein S21	*RPS21*
ENSSSCG00000008829	182	0.80	0.0076	OCIA domain containing 2	*OCIAD2*
ENSSSCG00000026044	221	1.34	0.0005	farnesyl-diphosphate farnesyltransferase 1	*FDFT1*
ENSSSCG00000006238	84	2.11	0.0007	cytochrome P450 family 7 subfamily A member 1	*CYP7A1*
**Rapeseed Oil vs. Beef Tallow**					
ENSSSCG00000006238	100	−2.63	4 × 10^-8^	cytochrome P450 family 7 subfamily A member 1	*CYP7A1*
ENSSSCG00000028821	27	−2.14	1.6 × 10^5^	SAS-6 centriolar assembly protein	*SASS6*
ENSSSCG00000004586	19	−2.02	6.5 × 10^5^	family with sequence similarity 81 member A	*FAM81A*
ENSSSCG00000033822	110	−1.91	6.6 × 10^5^	thyroid hormone responsive	*THRSP*
ENSSSCG00000026044	252	−1.80	1.6 × 10^5^	farnesyl-diphosphate farnesyltransferase 1	*FDFT1*
ENSSSCG00000006719	94	−1.41	1.6 × 10^7^	hydroxy-delta-5-steroid dehydrogenase, 3 β- and steroid delta-isomerase 1	*HSD3B1*
ENSSSCG00000006040	186	−1.20	2.6 × 10^6^	dihydropyrimidinase	*DPYS*
ENSSSCG00000001849	153	−0.89	1.7 × 10^4^	alanyl aminopeptidase, membrane	*ANPEP*
ENSSSCG00000039388	448	−0.84	8.3 × 10^5^		
ENSSSCG00000023686	5283	−0.78	3.4 × 10^4^	transthyretin	*TTR*
ENSSSCG00000015106	124	1.23	1.5 × 10^4^	hypoxia up-regulated 1	*HYOU1*
ENSSSCG00000032381	218	1.35	3.4 × 10^4^	lipopolysaccharide induced TNF factor	*LITAF*
ENSSSCG00000011297	222	1.36	6.6 × 10^5^	abhydrolase domain containing 5, lysophosphatidic acid acyltransferase	*ABHD5*
ENSSSCG00000015140	293	1.38	8.7 × 10^9^	heat shock protein family A (Hsp70) member 8	*HSPA8*
ENSSSCG00000030095	78	1.49	2.4 × 10^4^	zinc finger and BTB domain containing 16	*ZBTB16*
ENSSSCG00000020754	32	1.51	6.5 × 10^5^	RNA polymerase I subunit G	*CD3EAP*
ENSSSCG00000005601	1136	1.54	1.8 × 10^13^	heat shock protein family A (Hsp70) member 5	*HSPA5*
ENSSSCG00000022126	97	1.73	4.0 × 10^13^	epidermal growth factor receptor	*EGFR*
ENSSSCG00000010686	369	1.79	8.3 × 10^6^	BAG cochaperone 3	*BAG3*
ENSSSCG00000035058	43	1.82	6.3 × 10^6^	phosphotyrosine interaction domain containing 1	*PID1*
**Rapeseed Oil vs. Coconut Oil**					
ENSSSCG00000023331	67	−1.35	0.0000	ubiquitin like 5	*UBL5*
ENSSSCG00000007710	75	−1.14	0.0033	MLX interacting protein lik	*MLXIPL*
ENSSSCG00000027926	94	−1.14	0.0062	formimidoyltransferase cyclodeaminase	*FTCD*
ENSSSCG00000003302	193	−1.01	0.0004	Lysoplasmalogenase	***TMEM86B***
ENSSSCG00000031881	48	−1.01	0.0003	CDC42 small effector 1	*CDC42SE1*
ENSSSCG00000035790	110	−0.93	0.0048	BTG anti-proliferation factor 1	*BTG1*
ENSSSCG00000006719	105	−0.92	0.0013	hydroxy-delta-5-steroid dehydrogenase, 3 β- and steroid delta-isomerase 1	***HSD3B1***
ENSSSCG00000031302	46	−0.91	0.0015	C-terminal binding protein 1	*CTBP1*
ENSSSCG00000010627	45	−0.88	0.0064	programmed cell death 4	*PDCD4*
ENSSSCG00000013514	225	−0.73	0.0064	LRRCT domain-containing protein	***PLIN5***
ENSSSCG00000001849	163	−0.65	0.0062	alanyl aminopeptidase, membrane	*ANPEP*
ENSSSCG00000014855	870	−0.62	0.0002	ribosomal protein S3	*RPS3*
ENSSSCG00000011000	253	0.86	0.0013		
ENSSSCG00000022126	69	0.94	0.0007	epidermal growth factor receptor	*EGFR*
ENSSSCG00000039147	108	0.96	0.0014		
ENSSSCG00000015140	278	1.14	0.0000	heat shock protein family A (Hsp70) member 8	*HSPA8*
ENSSSCG00000004093	290	1.14	0.0000	iodotyrosine deiodinase	*IYD*
ENSSSCG00000010686	339	1.49	0.0000	BAG cochaperone 3	*BAG3*
ENSSSCG00000011437	132	1.50	0.0005	5′-aminolevulinate synthase 1	*ALAS1*
ENSSSCG00000024596	69	1.62	0.0051	nocturnin	*NOCT*

Bold genes were annotated with the Uniprot database.

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
