# Peer review of "3′quant mRNA-Seq of Porcine Liver Reveals Alterations in UPR, Acute Phase Response, and Cholesterol and Bile Acid Metabolism in Response to Different Dietary Fats"

_genes, 2020, doi:10.3390/genes11091087_

Round 1
Reviewer 1 Report
The research is very interesting but I believe that more attention to the Material and Methods section should be provide. The composition of the groups of animals, and so the provenience of the samples, should be described in details.
Line 70: “(12 gilts and twelve barrows)”, or number or word. Please change that.
Line 72: “a few samples were excluded”, from 24 samples 16 samples, did the authors excluded The authors used for the research samples from gilts and barrows but it’s not clear how the groups were composed in terms of number of gilts/barrows. It would be interesting to know in details how the research has been carried out. There might be also differences in the food intake between gilts and barrows, did the authors observe something?
Line 114-115: The authors already explained how the three different diets are composed. It might be redundant.

Author Response
Reviewer 1
Dear Reviewer, We would like to thank you for all your comments. We have improved the manuscript according to them, all errors have been corrected, all the recommendations mentioned in the review have been introduced into the manuscript. Moreover, the manuscript has been corrected by MDPI English editing service.
Line 70: “(12 gilts and twelve barrows)”, or number or word. Please change that. corrected
Line 72: “a few samples were excluded”, from 24 samples 16 samples, did the authors excluded. The authors used for the research samples from gilts and barrows but it’s not clear how the groups were composed in terms of number of gilts/barrows.
From 24 animals from nutritional experiment (the whole experiment included analyses of carcass parameters, growth traits, backfat quality traits and were described elsewhere) , 18 samples were collected for RNA sequencing. After RNA-seq we have excluded additional two samples due to low quality of sequencing. This have caused sex imbalance in rapeseed oil and beef tallow group, although males used in the study were castrated (barrows). To assess the effect of the sex on our results we performed a comparison between females and males with DEseq2 software, regardless of dietary group. We observed only eleven differentially expressed genes between female and males, all of them were located on sex chromosomes. None of the genes differentially expressed between males and females were present among the genes identified as a differentially expressed between dietary groups. Furthermore, expression of none of the genes analysed by qPCR was affected by sex. Thus, we concluded that the effect of sex is negligible in our study and that it is better to exclude low quality samples. Now, we have included the results of the comparison between males and females as a supplementary table 4 and included additional explanation in the results section – lines 160-165. The composition of groups was included to material and methods section, lines 72-75.
There might be also differences in the food intake between gilts and barrows, did the authors observe something?
Since the animals obtained restricted amounts of feed, we have wrongly used term “feed intake” instead of “average feed utilization “, now we have corrected this mistake, line-127.
All animals in the experiment (both gilts and barrows) were fed the same way with restricted feed amounts. Each animal was housed in an individual cage and received own daily weighed ration of feed, to meet their requirements. The portion was administered in two parts (at 7 a.m and 2 p.m.). The fact that the daily ration of the feed was adjusted to the body weight of the animals and divided into two portions, made it possible to avoid leaving unconsumed feed. Therefore, the feed intake was the same and we assume no effect of the amount of consumed feed on the results of the experiment.
Line 114-115: The authors already explained how the three different diets are composed. It might be redundant. The sentence was removed

Reviewer 2 Report
3`quant mRNA-seq of porcine liver reveals alterations 2 in UPR, acute phase response and cholesterol and bile 3 acid metabolism in response to different dietary fats.
Major concerns
- The diet formulation should be presented, particularly, the calculated ME. Since rapeseed oil, beef tallow and coconut comprise quiet different fatty acid compositions, which vary with carbon lengths and thus contain different calories in energy. So, if ME are not different among the diets, how the authors concluded the results being caused by the specific fatty acid species of the diets.
- Some molecular and physiological results should be included to potentiate the results, for example, liver or circulating inflammation status such as IL-6 and TNFa levels by Western blot or ELISA analysis. Also, some functional/pathological studies such as bile acid species and content in the intestine and liver and histological examination may be analyzed.
- In Fig 7, the authors validated the differentiation geneexpression by qRT-PCR. Since the results were derived from transcriptomic analysis, the authors may validate some of results at protein levels using Western blot or ELISA analysis.
- In, table 2, the authors had bettern to present the whole name of genes to facilitate the readers to know their functions.

Author Response
Reviewer 2
Dear Reviewer, We would like to thank you for all your comments. We have improved the manuscript according to them, all errors have been corrected, all the recommendations mentioned in the review have been introduced into the manuscript. Moreover, the manuscript has been corrected by MDPI English editing service.
The diet formulation should be presented, particularly, the calculated ME. Since rapeseed oil, beef tallow and coconut comprise quiet different fatty acid compositions, which vary with carbon lengths and thus contain different calories in energy. So, if ME are not different among the diets, how the authors concluded the results being caused by the specific fatty acid species of the diets.
The composition, nutritive value and metabolisable energy of used feed mixtures were already presented [10]. In order to balance the recipe of the mixtures, a computer program is used based on the Polish tables for the chemical composition and nutritional value of feeds. According to these data, the used fats’s energy value for swine were similar (rapeseed oil - 36.2 MJ/kg, beef tallow - 35.3 MJ/kg, coconut oil - 36.7 MJ/kg). After using a 3% content of these components, mixtures with a similar energy concentration were obtained (R- 13.4 MJ/kg, B - 13.3 MJ/kg, C - 13.4 MJ/kg). From the point of view of pig feeding practice, these differences are negligible. Assuming that the energy value of the three fats is similar, we believe that the differences observed in the experimental results may be due to the non-caloric properties of these fats, but due to differences in fatty acids composition and their digestibility route as well as the health properties of some fatty acids.
Some molecular and physiological results should be included to potentiate the results, for example, liver or circulating inflammation status such as IL-6 and TNFa levels by Western blot or ELISA analysis. Also, some functional/pathological studies such as bile acid species and content in the intestine and liver and histological examination may be analyzed.
In Fig 7, the authors validated the differentiation gene expression by qRT-PCR. Since the results were derived from transcriptomic analysis, the authors may validate some of results at protein levels using Western blot or ELISA analysis.
We share the reviewer's concerns, however this project was designed as a typical transcriptomic study, with the assumption that mRNA expression is a good proxy for protein expression .
Nevertheless, we have added to the text a few sentences of explanation: “Although our work provides a huge amount of new information on the impact of consuming different sources of fat on gene expression in the liver, we are aware that these studies should be extended with analyzes at the level of proteins and their functioning. Furthermore, bile acid species content in the intestine and liver analyses and histological examination of these organs would give a full view of changes introduced by consumption of different sources of fat” – lines 477-482. Additional sentence: ” Nevertheless, these studies should be extended with analyzes at the level of proteins and their functioning.” was also added to the abstract in order to underline limitation of the study – lines 31-32.
In, table 2, the authors had bettern to present the whole name of genes to facilitate the readers to know their functions. Gene descriptions has been added to the table.

Round 2
Reviewer 1 Report
In my opinion the paper is now suitable for the publication.
Author Response
Dear reviewer,
Thank you again for your comments!
Best regards,
Maria Oczkowicz
Reviewer 2 Report
In line 83, the resultant diets with similar ME should be added in (R 13.4, B 13.3 , C 13.4 MJ/kg feed).
Author Response
Dear reviewer,
Thank you again for your comments,
The information about ME has been added -line 77,
Best regards,
Maria Oczkowicz